# Loss of Homeostatic Microglia Signature in Prion Diseases

**DOI:** 10.3390/cells11192948

**Published:** 2022-09-21

**Authors:** Yue Wang, Kristin Hartmann, Edda Thies, Behnam Mohammadi, Hermann Altmeppen, Diego Sepulveda-Falla, Markus Glatzel, Susanne Krasemann

**Affiliations:** Institute of Neuropathology, University Medical Center Hamburg-Eppendorf (UKE), 20251 Hamburg, Germany

**Keywords:** prion diseases, microglia, homeostatic microglia, Creutzfeldt-Jakob disease, neuro-inflammation, neurodegenerative diseases, prion protein, astrocytes

## Abstract

Prion diseases are neurodegenerative diseases that affect humans and animals. They are always fatal and, to date, no treatment exists. The hallmark of prion disease pathophysiology is the misfolding of an endogenous protein, the cellular prion protein (PrP^C^), into its disease-associated isoform PrP^Sc^. Besides the aggregation and deposition of misfolded PrP^Sc^, prion diseases are characterized by spongiform lesions and the activation of astrocytes and microglia. Microglia are the innate immune cells of the brain. Activated microglia and astrocytes represent a common pathological feature in neurodegenerative disorders. The role of activated microglia has already been studied in prion disease mouse models; however, it is still not fully clear how they contribute to disease progression. Moreover, the role of microglia in human prion diseases has not been thoroughly investigated thus far, and specific molecular pathways are still undetermined. Here, we review the current knowledge on the different roles of microglia in prion pathophysiology. We discuss microglia markers that are also dysregulated in other neurodegenerative diseases including microglia homeostasis markers. Data on murine and human brain tissues show that microglia are highly dysregulated in prion diseases. We highlight here that the loss of homeostatic markers may especially stand out.

## 1. Introduction

Prion diseases, also known as transmissible spongiform encephalopathies, are neurodegenerative, progressive disorders. They are always fatal and, to date, no therapeutic or disease-modifying strategies exist. Prion diseases affect humans and animals alike and, occasionally, may even be transmitted between different species. The animal prion disorders include scrapie in sheep and goats, bovine spongiform encephalopathy (BSE; also known as “mad cow disease”) in cattle, and chronic wasting disease in deer and elk [1,2,3]. The human diseases include Creutzfeldt–Jakob disease (CJD), Gerstmann–Sträussler–Scheinker Syndrome, Kuru, and Fatal Familial Insomnia [4,5,6,7,8,9,10]. The most common human prion disease is sporadic CJD (sCJD) with an incidence of 1–2 per million people per year [11]. Several sCJD subtypes have been defined, which may differ in clinical presentation. The molecular basis for this phenotypic variability is defined by the distinct biochemical properties of the prion protein and an allelic variation in codon 129 (methionine or valine) of the prion protein gene (*PRNP*) [12,13]. This polymorphism also modulates the degree of disease susceptibility, e.g., for BSE, since BSE has been transmitted to humans in a number of cases, mainly in 129 MM carriers during the BSE crisis, and gave rise to a subtype of CJD, which was termed variant CJD [14,15]. However, variants of *PRNP* were also identified as a key risk factor for sCJD, besides the recent identification of novel risk loci [16,17]. Upon adaptive passaging, prion diseases can also be transmitted to small animal models such as hamsters and mice, and such adapted prion strains and animal models have been intensively used to study the pathophysiology of the disease. Prion diseases are characterized by the conformational conversion of the endogenous cellular prion protein PrP^C^ into the disease-associated protein isoform PrP^Sc^, which is key to prion formation and disease progression [18] (Figure 1A). PrP^C^ is a glycosylphosphatidylinositol (GPI)-anchored protein that is attached to the outer leaflet of the plasma membrane [19]. Misfolding of this cellular isoform leads to the β-sheet-enriched and protease-resistant disease conformer PrP^Sc^ [20,21]. Since PrP^C^ is the essential substrate for the disease-associated PrP^Sc^-templated misfolding cascade, PrP-knockout mice are resistant to prion infection and disease [22]. Besides the accumulation of aggregated PrP^Sc^ in the brain, prion diseases are characterized by neuronal loss, spongiform lesions, and the widespread activation of astrocytes and microglia (Figure 1B).

To date, the mechanisms leading to neurotoxicity and neuronal loss are only partially understood, yet cellular levels of PrP^C^ seem to critically determine neurotoxicity [23]. Toxic signaling of misfolded PrP^Sc^ via cellular receptors (with PrP^C^ possibly being one of them) on the neuronal membrane may lead to direct synapto- and neurotoxicity [24,25]; however, non-neuron autonomous pathways may contribute as well. The view that microglia might be a critical component of prion disease pathophysiology was already noted decades ago [26,27,28]. Here, we review the current knowledge of microglia and their involvement in prion disease pathophysiology. We show staining examples of different microglial markers in prion mouse models and human sCJD. In agreement with previous studies of our and other research groups, microglia show progressive dysregulation during the course of prion disease in a region-dependent manner.

### 1.1. Microglia Are Dysregulated and May Have a Dual Role in Prion Diseases

Small rodents can be infected with prions from various natural sources (i.e., isolated from animals such as sheep that succumbed to prion disease and then serially passaged in mice). These experiments led to the generation of different mouse-adapted prion variants, so-called strains (such as 22L, RML, ME7, etc.) with distinct neuropathological features and disease kinetics [29,30]. Hamster and mouse models faithfully recapitulate all aspects of prion disease with generation and deposition of misfolded PrP^Sc^, astrogliosis, and microgliosis, and also including behavioral changes and a shortened life span. Therefore, prion-infected mice represent a versatile tool to study and understand prion biology [31]. The use of these rodent models also led to the early discovery that microglia are activated in prion diseases with the mounting of an atypical inflammatory response [32,33,34].

The clinical course of prion disease in mice is accurately predictable, suggesting that prion pathogenesis is driven by precisely timed molecular events [35]. In the RML infection mouse model of prion diseases, the incubation time to terminal disease is about 150 days ± 5 days (Figure 2) [36]. While infectious prions can be detected in the brains of intracerebrally infected animals at about 30 dpi, misfolded PrP^Sc^ can be detected by Western blotting only after about 60 dpi. Symptomatic disease starts at around day 90, with mice displaying a stiff tail and a gradual loss of nest-building behavior. The latter deteriorates from the building of a proper nest via the unorganized stacking of nesting material to the complete absence of a nest, with the material tramped to the floor and soiled by excrement close to terminal disease. Progressive dysregulation of glia cells including microglia could also be demonstrated on the gene expression level using novel techniques [37,38]. Interestingly, while microglia expression profiles changed considerably of the course of disease in a region-dependent manner, neuronal phenotypes remained comparatively stable [37,38]. Microglia are the innate immune cells of the brain, yet in contrast to their counterparts in peripheral organs, the macrophages, they arise from primitive yolk sack macrophages that colonize the brain early during development [39,40]. In the adult brain, microglia replenish by a self-renewal process and local proliferation upon stimulation [41,42,43,44]. Different groups have independently shown that the appearance of reactive microglia starts before the onset of symptoms, can be detected at around day 70–80 in a region-dependent manner, and is visible mainly in the thalamus at this time point (Figure 2) [45,46,47]. In terminal prion disease, two changes in microglia are apparent: First, microglia morphology is changed from the ramified presentation typical in homeostasis to a bushy/reactive or even amoeboid morphology in a region-dependent manner (Figure 3A,B) [45,46,47]. The number of activated microglia is highest in the thalamus, a region where reactive microglia can already be detected at a preclinical time point (see Figure 2) [45,46,47]. Second, microglia proliferation leads to a significant increase in microglia numbers (Figure 3B). Although microglia numbers were significantly increased in all four brain regions at the terminal stage, the increase was highest in the thalamus [45,46,47]. The region-specific increase in microglia numbers during the course of a prion disease had already been shown with several mouse adapted prion strains [45,46,47] and may be aggravated by systemic infection [48,49]. Moreover, it has been shown that suppression of microglia proliferation in the clinical disease phase significantly prolonged survival in the mouse model [45], suggesting a detrimental role for this cell type at a late disease stage. In contrast, reducing microglia numbers by knocking out interleukin 34 (IL-34), an essential signaling molecule, led to augmented PrP^Sc^ deposition and shortened survival in a prion mouse model [50,51], while the inhibition of IL-34 resulted in reduced microglia proliferation [52]. Moreover, partial depletion of microglia in early stages of the disease was reported to enhance the accumulation of prions in the brain and accelerate the onset of clinical disease [46]. Novel data on the complete depletion of brain microglia have shown that mice displayed an accelerated disease course, but did not show an enhanced accumulation of PrP^Sc^ [53]. These controversial outcomes highlight a complex dual role of microglia also observed in other brain diseases [54,55,56] and might be attributed to the fact that the total removal of microglia might be harmful, whereas targeting specific microglia activation states later in disease might prolong survival time. Interestingly, novel data have also shown that M-CSF/CSF1R signaling is upregulated in prion infection [57]. Work on prion-infected organotypic brain-slice cultures support the prion clearing and neuroprotective functions of microglia [50]. This is in line with findings showing that microglia isolated from the brains of prion-infected mice are infectious [58]. Moreover, microglia are directly activated by PrP^Sc^ in vitro [33,59]; however, such causalities are difficult to prove in vivo. Novel research in mice that exclusively produce PrP^Sc^ in neurons or in astrocytes shows that neither drive the dysregulation of glia cells on their own [60]. These findings warrant more research about the putative cross-talk of the different brain cell types in disease.

As mentioned above, microglia in the healthy brain display a homeostatic morphology characterized by a small cell body and long branching processes which are constantly surveying their environment [61]. Moreover, microglia are involved in several mechanisms regulating brain development and plasticity. They actively shape neuronal connectivity by the removal of excess/neglected synapses, which are formed during development [62,63,64]. For this, microglia have been shown to directly contact tagged pre- and postsynaptic structures and remove them by phagocytosis. The targeting of unwanted elements for removal involves complement receptors and adaptor proteins, among other signals [62,64].

Microglia can react to alterations in the brain environment or the presence of threats to its cellular integrity e.g., invading pathogens and aggregated proteins, but also apoptotic cells or cell debris, by mounting an inflammatory response to restore homeostasis. The latter enables microglia to increase their phagocytic capacity to remove unwanted structures and to release proinflammatory mediators including cytokines and chemokines [65]. Activated microglia represent a common feature of neurodegenerative diseases [54,66,67]. Activation can be identified by a combination of morphological and immuno-phenotypic changes [68]. Several proinflammatory cytokines are upregulated in the brain during the prion disease course including TNF α, IL 1α, and C1qa [46,69,70,71,72,73]. The same outcome was recently published after region-specific bulk transcriptomic analyses of brains infected with two different prion strains [74]. Although these strains target different cell populations in the brain, the regional glia response was comparable. This is in line with further research using three different prion strains, RML, 22L, and ME7 which produce different patterns of PrP^Sc^ deposition. Despite these obvious differences, highly similar expression patterns in neuroinflammatory genes and glia response genes were measured in all three mouse models [75].

Since pro-inflammatory mediators are upregulated slightly before the onset of symptoms, several studies were conducted to investigate the impact of manipulating microglia receptor abundance or cytokine release, and yielded a somewhat mixed outcome. In line with the proposed capacity of microglia to phagocytose PrP^Sc^, the knockout of C-X-C chemokine receptor type 3 (CXCR3) led to accelerated PrP^Sc^ accumulation and increased prion infectivity titers, but prolonged survival in the mouse model [76]. Moreover, these mice developed excessive astrocytosis. The knockout of the cluster of differentiation CD14 also led to a prolonged survival with enhanced microglia activation in two different mouse models [77]. The NLRP3 inflammasome is a multi-molecular complex which can sense heterogeneous pathogen-associated molecular patterns (PAMPs) culminating in the activation of caspase 1 and the release of interleukin IL 1β. It has been shown to play a fundamental role in Alzheimer’s disease (AD) pathophysiology [78,79]. In contrast to AD, the knockout of NLRP3 or the adaptor protein ASC does not influence the prion disease course [80]. Interestingly, retroviral infection preceding prion infection in mice led to a stimulation of microglia at early disease time points with a reduction in infectious prions [47], while another stimulator of inflammation, the bacterial lipopolysaccharide (LPS), could not further stimulate their PrP^Sc^-degrading function during disease progression [81]. In addition to the upregulation of PAMPs in prion diseases, the transcription of several genes encoding damage-associated molecular pattern (DAMP) proteins and receptors such as Toll-like receptors or proteins of the complement cascade are also increased in the brains of prion-infected mice [70,82]. Interestingly, depletion of any of the DAMP receptor genes *Tlr2*, *C3ar1*, and *C5ar1* in a prion mouse model only led to a slightly increased survival in the *Tlr2*-model, while *C3ar1*, and *C5ar1* did not influence prion disease course [82].

Since these studies did not lead to conclusive data about the role of microglia and their putative PrP^Sc^-reducing function, another line of experiments was performed to address the impact of “eat me”-signals on neurons or “eat-me”-receptors on microglia on neuronal survival in prion pathophysiology [83]. These experiments were fueled by the notion that the milk fat globule-EGF factor 8 protein (MFGE8) produced by astrocytes might target apoptotic bodies via binding to their surface-exposed phosphatidylserine for their clearance by interaction with the MFGE8 receptor on microglia in prion disease. Accordingly, the knockout of MFGE8 accelerated disease but this was restricted to certain mouse models [84]. However, neither the “eat me”-protein developmental endothelial locus-1 nor the “don’t eat me”-receptor signal regulatory protein SIRPa, nor the macrophage scavenger receptor 1 (Msr1) influence the prion disease course, since their depletion had no effect on disease pathophysiology [85,86,87].

### 1.2. Homeostatic Microglia: Guardians of the Physiological State

For decades, microglia research was focused on the inflammatory context and profile. Only recent research has brought into the focus the homeostatic functions of microglia, which—if lost—might be a driver of neurodegeneration in disease [55,88,89,90,91]. However, the identification of microglia-specific profiles was complicated by the lack of understanding of whether and how brain-resident microglia functionally differ from peripheral myeloid cells which might enter the brain in certain instances [92]. Only recently were the molecular and functional characteristics of murine bona fide homeostatic microglia identified by different groups using a set of novel methodologies such as quantitative proteomics or RNA sequencing [88,93,94]. Thus, a unique transcriptional expression signature of homeostatic microglia was identified which allowed for their differentiation from peripheral myeloid cells. Homeostatic microglia are characterized by the expression of specific markers such as *Tmem119*, *Hexb*, *Gpr34*, *P2ry12*, *Olfml3*, and *Tgfbr1*. Several of these genes are also expressed on human microglia, including the P2Y purinoceptor 12 (*P2RY12*) [95] and the transmembrane protein *TMEM119* [96]. The development of robust tools including microglia-specific antibodies followed soon after.

We recently showed that the abundance of homeostatic microglia proteins such as TMEM119 and P2RY12 is significantly reduced in terminal prion disease (Figure 4) [47,73,97]. In the healthy brain, microglia show a ramified morphology with a small cell body and thin processes that are especially visible after staining with P2RY12 [55,88,89,90,91]. In contrast, both markers are affected in terminal prion disease. While the homeostatic markers are still abundant in the hippocampus, this signature is almost completely lost in the thalamus, an effect which is especially severe for P2RY12 (Figure 4) [47,73,97].

The loss of homeostatic microglial proteins is visible in immunofluorescence double-staining of IBA1 and TMEM119 in terminally sick mice and mock-infected healthy control brains, and shows the reciprocal regulation of both proteins in prion disease (Figure 5A) [55,73,88,89,90,91]. Both markers co-localize, highlighting the high specificity for microglia and the ramified morphology with fine processes that are mainly stained by TMEM119 in the healthy brain [55,88,89,90,91]. As previously shown, TMEM119-positive processes are highly reduced at the terminal stage of prion disease in the hippocampus, and almost completely lost in the thalamus during terminal prion disease (Figure 5) [73]. Interestingly, these findings on the loss of homeostatic microglial proteins in terminal prion disease are in contrast to findings on the RNA expression of the corresponding genes, since *P2ry12* RNA expression has been shown to be relatively upregulated [98,99]. On one hand, this might be due to the usage of different prion strains, disease stages, or tissue sampling. On the other hand, discrepancies in RNA levels as compared to actual protein expression in microglia activation have been noted before (see also upcoming white paper on microglia: Defining Microglial States and Nomenclature: A Roadmap to 2030 https://papers.ssrn.com/sol3/papers.cfm?abstract_id=4065080 (accessed on 10 September 2022)) [100]. Thus, further experiments including mass spectrometry analyses of microglial protein expression at different disease stages might help to solve this controversy. The homeostatic microglia signature is also downregulated in other neurodegenerative diseases. We and others have identified a disease-specific microglia signature that is commonly dysregulated in brain diseases including AD [55,101]. The transcriptional phenotypes of microglia in neurodegenerative disease (MGnD) or disease-associated microglia (DAM) highlight several proteins that actively influence disease progression. These include apolipoprotein E (ApoE) and triggering receptor expressed on myeloid cells-2 (TREM2). Both proteins are upregulated in prion disease and affect the expression of pro-inflammatory proteins during the disease course in murine models. However, their depletion did not change survival times [102,103,104]. However, the deletion of ApoE might influence the course of prion disease including the exacerbation of prion pathology, a dysregulated microglial phenotype, and impaired clearance of PrP^Sc^ and dying neurons by microglia [104]. The microglia signature in prion diseases also differs from that seen in other neurodegenerative diseases [104]. Fittingly, we found that the loss of the homeostatic signature is especially severe in prion diseases compared to other neurodegenerative conditions [47,55].

Another characteristic of prion diseases is the widespread and severe reactive astrogliosis [105]. Astrocyte activation was noted as a specific hallmark of prion diseases, with significant upregulation of the glial fibrillary acidic protein (GFAP). Thus, the impact of its knockout on disease pathophysiology was tested several years ago [102,106]. Interestingly, GFAP-knockout did not influence prion disease outcome. GFAP is highly upregulated in astrocytes upon disease [73,75,107]. Microglia and astrocytes keep close contacts in the healthy brain but also in terminal prion disease, although both cell types are considerably dysregulated (Figure 5B) [73]. The functional consequences of these contacts in health or disease are not yet clear.

Recent investigations on astrocyte dysregulation in neurodegenerative diseases have provided substantial evidence that a proinflammatory microglial cytokine cocktail containing TNF α, IL 1α, and C1qa reprograms a subset of astrocytes to change their expression profile and phenotype, thus becoming neurotoxic (designated A1 astrocytes) [108]. We recently showed that astrocytes in murine and human prion diseases express the A1-signature proteins complement 3 and GBP2 [73]. However, a typical A1 profile could not be identified. In contrast, astrocytes in prion disease showed a mixed phenotype with an expression of both A1 and A2 proteins that is unique in prion disorders and differs from other neurodegenerative diseases [73]. Interestingly, novel investigations in mouse models after microglia depletion have shown that this prion disease-specific astrocyte signature develops without the proinflammatory stimulation provided by microglia [98].

### 1.3. Human Microglia and Their Regional Heterogeneity

For decades it has been noted, mainly based on morphological data from immunohistochemical staining, that the microglia population within the brain is not uniform. Over the last ten years, microglia heterogeneity has been studied in murine models, determining that microglia have distinct region-dependent transcriptional identities and that they age in a regionally variable manner [109,110]. Recent advances in single-cell sequencing have shown the diversity and regional heterogeneity of murine brain microglia in more detail [111]. Interestingly, these data show that microglia diversity is the highest in the developing, aged, or injured brain. Using a mouse model for multiple sclerosis (MS) and human MS brain tissues, the authors found diverse activated microglia subpopulations in demyelinating lesions [111]. Other groups complemented these data by combining advanced mass spectrometry methods and single-cell RNA sequencing, revealing comparable data from human brain tissue [100,112].

Although a variety of new molecular tools, such as nuclear sequencing, are now available to study expression profiles on a single-cell basis in the human brain in health and disease [113], the transmissible nature of the fatal yet untreatable prion disorders and their respective biosafety issues limit their use in the study of human prion disease. However, bulk expression analysis in two brain regions in sCJD displayed regionally distinct inflammatory profiles [114].

We have previously shown that the TMEM119 protein level is significantly reduced in the brains of sCJD patients (Figure 6A) [97]. See Table 1 for an overview of the human tissues displayed in this review. Despite high amounts of PrP^Sc^ deposition in the frontal cortex and cerebellum in a very distinct pattern, the activation of microglia shown by IBA1 is evenly high throughout these entire regions. Prion disease-associated astrogliosis is also prominent in human prion diseases as shown by the upregulation of GFAP and YKL-40 (or Chitinase-3-like protein 1; CHI3L1) (Figure 6B) [73,107]. However, while a GFAP increase is also found in other neurodegenerative diseases such as AD [97], YKL-40 upregulation is more specific for human CJD and might serve as a diagnostic marker [107]. As another marker for diagnostic purposes, soluble TREM2 (sTREM2) in cerebrospinal fluid might emerge [115]. Although TREM2 is upregulated in murine and human prion diseases, it might not play a role in disease modification [103,115]. However, while it may not be a therapeutic target in prion disease, it might well serve as a diagnostic tool to discriminate prion disease from other (rapidly progressing) neurodegenerative diseases such as AD in patients. The microglia signature of sCJD patients differ from those of rapid AD, with significant downregulation of the microglia homeostatic marker TMEM119 (Figure 7) [97]. Moreover, the activation marker cluster of differentiation CD68 was increased in the frontal cortex of a patient with sCJD compared to AD patients (Figure 7) [97]. Microglia are activated in both diseases. Interestingly, the loss of the homeostatic marker seems more severe in sCJD compared to AD [97]. A significant increase in the abundance of CD68 compared to healthy controls, in a region-specific manner and dependent on the sCJD subtype, has already been shown in an independent study [116]. These data are in line with recent findings that patients with AD only show a moderate loss of TMEM119 [95,97]. However, immunohistochemical analyses and subsequent quantifications in a much bigger set of patients with sCJD and AD and non-demented control patients are warranted to determine disease and also to identify sCJD subtype-specific microglia dysregulation profiles.

As already determined in the murine models in more detail (see above), the regional differences in inflammation-associated expression changes are also beginning to be investigated and described in the frontal cortex and cerebellum of patients with sCJD [114]. Neuronal loss shows considerable variation between various regions of the brain within a CJD-diseased individual and between CJD patients. However, the cortex areas and cerebellum are often severely affected in sCJD, with a relative sparing of the hippocampus and subcortical grey matter [117,118]. Thus, it will be of the highest interest to investigate, if other, less-affected brain regions will show distinct inflammatory profiles in human prion diseases. Although the deposition patterns of misfolded PrP^Sc^ are diverse in murine models of prion disease, the heterogeneity of PrP^Sc^ deposition patterns is higher in human prion diseases and comprises synaptic, perivacuolar, plaque-like, kuru-plaque, florid-plaque, punctuate, perineuronal, and intraneuronal patterns, which might occur side-by-side or even overlapping in the same patient [119,120]. If and how these protein deposits directly influence the inflammatory profiles in individual brain regions in human prion diseases is currently not clear.

A more systematic investigation including different brain regions and a set of different techniques will certainly provide a better picture about region-dependent microglial (and astrocytic) signature shifts and their respective pathophysiological consequences in the future.

## 2. Conclusions

Microglia are dysregulated in neurodegenerative diseases. However, it is still not clear how microglia contribute to disease pathophysiology in detail: Is it the gain of detrimental proinflammatory functions? Or is it rather the loss of protective homeostatic support towards neuronal and other brain cell types? Or could it be a combination of both? The current data support the hypothesis that microglia activation at an early disease phase is beneficial and contributes to the removal of PrP^Sc^, whereas prolonged pro-inflammatory signaling and increased proliferation at later disease stages are harmful and accelerate neuronal decay. Thus, potential future therapeutic interventions targeting microglia in prion diseases certainly need to be thoroughly timed, dosed, and balanced. Although prion diseases, in several pathomechanistic regards, can be considered as a “prototype” for neurodegenerative diseases characterized by protein misfolding, we and others have shown that microglia are exceptionally highly dysregulated in murine and human prion diseases. They display an upregulation of activation markers that might be distinct from those in AD. Moreover, in our recent experiments, the extensive loss of the homeostatic signature in prion diseases stands out remarkably [47,73,97]. We thus speculate that certain mechanisms of disease, especially regarding glial responses, might be highly distinct in prion diseases and contrast with other neurodegenerative diseases such as Alzheimer’s disease.

## Figures and Tables

**Figure 1 cells-11-02948-f001:**
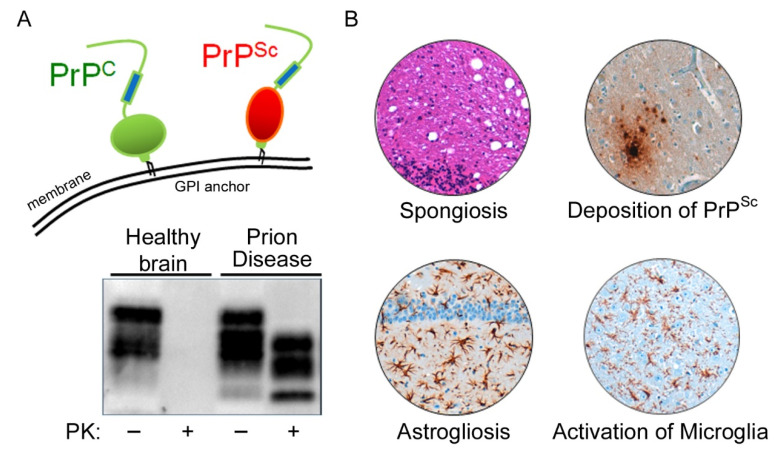
**The cellular prion protein is misfolded in prion diseases.** (**A**) A glycosylphosphatidylinositol (GPI) anchor tethers the cellular prion protein (PrP^C^; schematic in green) to the outer leaflet of the plasma membrane. In prion disease, PrP^C^ is conformationally converted into its disease-associated isoform PrP^Sc^ (schematic in red). Misfolded PrP^Sc^ is more stable against proteolytic digestion and can be detected in prion disease but not in healthy brain tissue after proteinase K (PK) digestion by Western blotting. (**B**) Prion disease histopathological hallmarks include spongiform changes (hematoxylin and eosin (H&E) staining, see vacuolation), deposition of aggregated protein (misfolded PrP^Sc^), astrogliosis (GFAP), and microglial activation (IBA1).

**Figure 2 cells-11-02948-f002:**
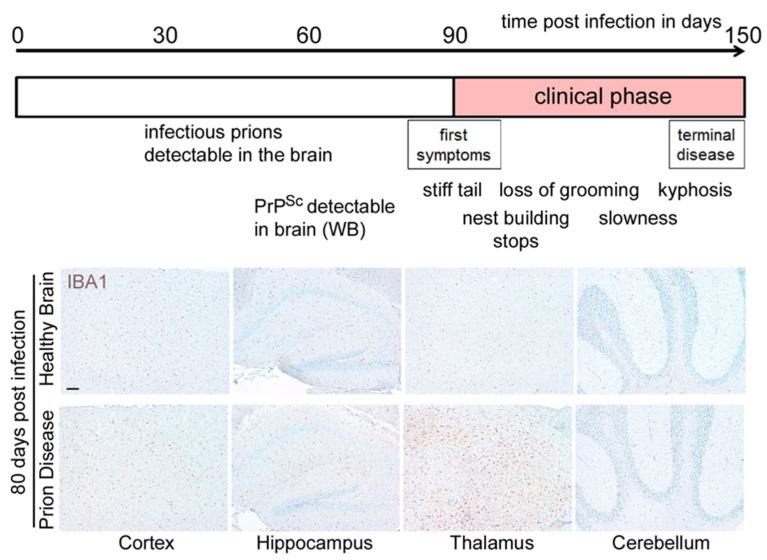
**Prion disease course in the mouse model.** Prion diseases are neurodegenerative diseases that can be easily experimentally transmitted by the inoculation of mouse-adapted prions (such as the Rocky Mountain Laboratory strain; RML inoculum) to the brains of mice. Disease onset and clinical course are relatively accurately predictable, eventually leading to terminal disease usually within 150 ± 5 days after high-dose intracerebral inoculation. RML 5.0 is a specific and very well-characterized passage of RML that was and is used in several laboratories [35]. After a latency period, in this model, prion infectivity is already detectable in the brain at 30 days post infection (dpi), and misfolded PrP^Sc^ can be detected by Western blot (WB) at around 60 dpi. Symptoms appear around 90–100 dpi. Lower panel: The pan-microglia/monocyte marker ionized calcium-binding adaptor molecule 1 (IBA1) is shown at 80 dpi in different brain regions in comparison to the brain of a mock-infected control mouse (healthy brain). Note the region-specific change towards a reactive microglia phenotype in the thalamus in prion disease during the preclinical stage before the onset of symptoms, including an increase in microglia numbers [45,46,47]. Scale bar: 100 µm.

**Figure 3 cells-11-02948-f003:**
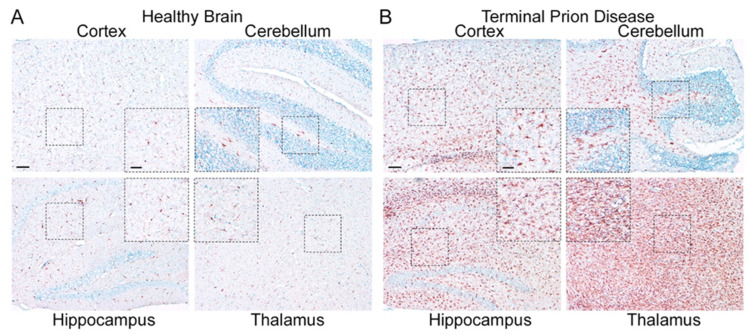
**Microglia are highly activated in terminal prion disease.** The pan-microglia/macrophage marker IBA1 is shown in mouse brain sections of (**A**) mock-infected healthy mice and (**B**) RML 5.0 prion-infected mice at a terminal disease state. (**A**) Microglia in the healthy brain show a ramified phenotype with a small soma and thin processes in all four regions displayed here (see magnified close-up). (**B**) During the terminal prion disease stage, microglia massively proliferate (see overview) and change their morphology towards bigger cell bodies and thicker arms (see close-ups) with a bushy appearance in the hippocampus and an amoeboid phenotype in the thalamus [45,46,47]. Scale bar: 100 µm; close-up: 50 µm.

**Figure 4 cells-11-02948-f004:**
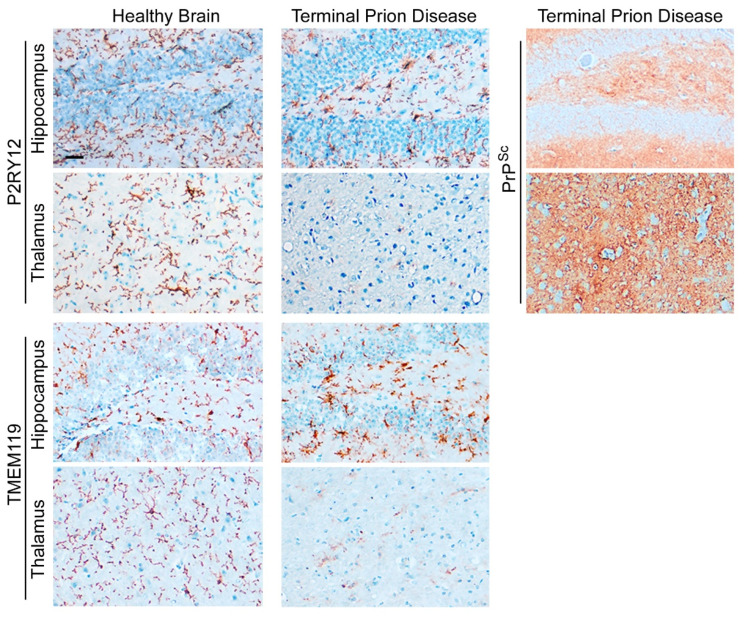
**Loss of microglial homeostatic phenotype in terminal prion disease.** The microglia-specific homeostasis markers P2RY12 and TMEM119 show a ramified morphology of microglia in the healthy brain [47,73,97]. In terminal prion disease, this homeostatic microglia signature is lost in a region-dependent manner [47,73,97]. While the activated microglia in the dentate gyrus of the hippocampus display a bushy morphology with thick and retracted processes and still express both markers, microglia in the thalamus (posterior complex) have almost completely lost the expression of the markers, especially P2RY12. In the RML-prion mouse model, the deposition of PrP^Sc^-specific staining is stronger in the thalamus than in the hippocampus in terminal prion disease. Scale bar: 25 µm.

**Figure 5 cells-11-02948-f005:**
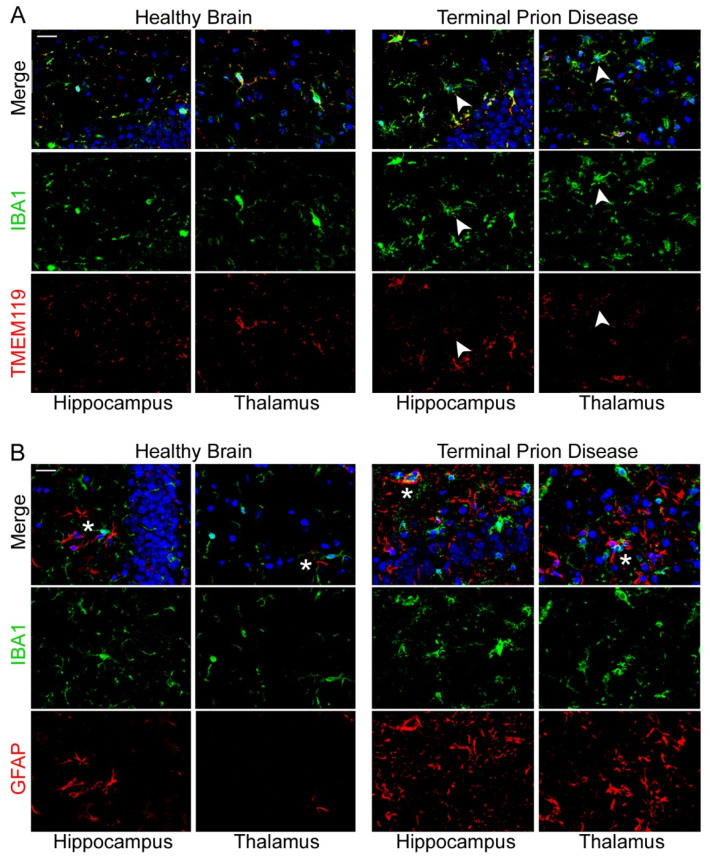
**Loss of microglia homeostatic phenotype and increase in gliosis upon terminal prion disease.** (**A**) IBA1 (green) and TMEM119 (red) co-localize in the hippocampus and thalamus in the healthy mouse brain, highlighting the ramified morphology with fine processes mainly stained by TMEM119 [47,73,97]. TMEM119 is highly reduced (white arrowhead) in the dentate gyrus of the hippocampus, and completely lost (white arrowhead) in the thalamus (posterior complex) in terminal prion disease [73]. Scale bar: 20 µm. (**B**) IBA1 (microglia/green) and GFAP (astrocytes/red) show intense dysregulation and are in close proximity (white asterisks) in the healthy brain and in terminal prion disease. (DAPI/nucleus in blue) [73]. Scale bar: 20 µm.

**Figure 6 cells-11-02948-f006:**
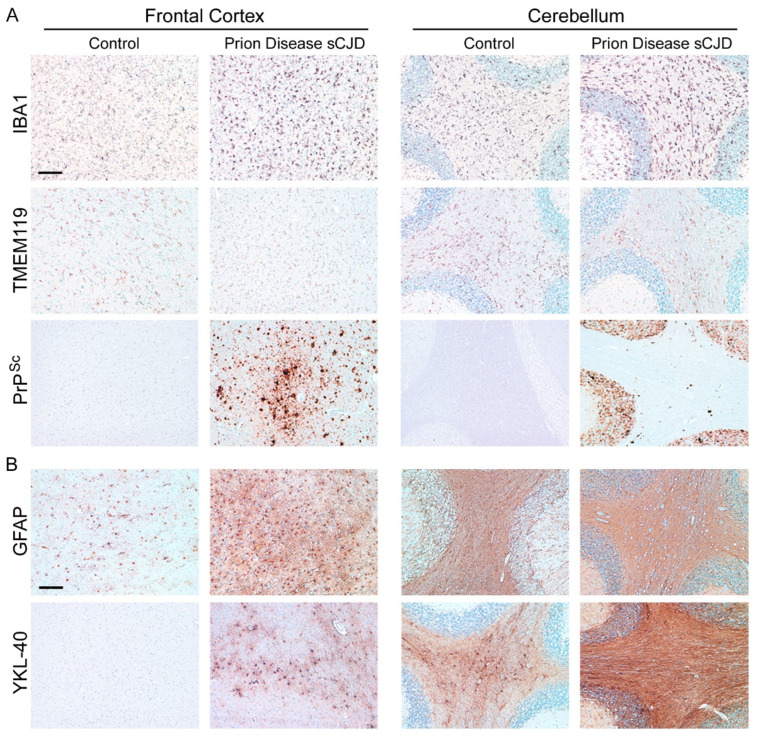
**Glia cells are highly dysregulated in human sCJD.** Human brain tissues are usually fixed in 4% buffered formalin. To inactivate prion infectivity, tissues should be incubated in 98% formic acid for 1.5 h. Tissues from healthy human controls should be similarly treated with formic acid, to enable identical staining conditions. Frontal cortex and cerebellum brain sections from sCJD patients and control patients without neurologic disease are shown for (**A**) the pan-microglia/macrophage marker IBA1 and homeostatic microglia marker TMEM119 [97]. Note that while IBA1+ microglia are increased in sCJD staining, the intensity of TMEM119 staining is reduced compared to an age-matched non-demented control patient [97]. Deposits of misfolded PrP^Sc^ (sCJD) are also shown. Scale bar: 200 µm. (**B**) Besides microglia, astrocytes are also highly dysregulated in prion diseases such as sCJD [97,107], as displayed by the astrocyte markers GFAP and YKL-40 for an sCJD case and an age-matched control. Scale bar: 200 µm.

**Figure 7 cells-11-02948-f007:**
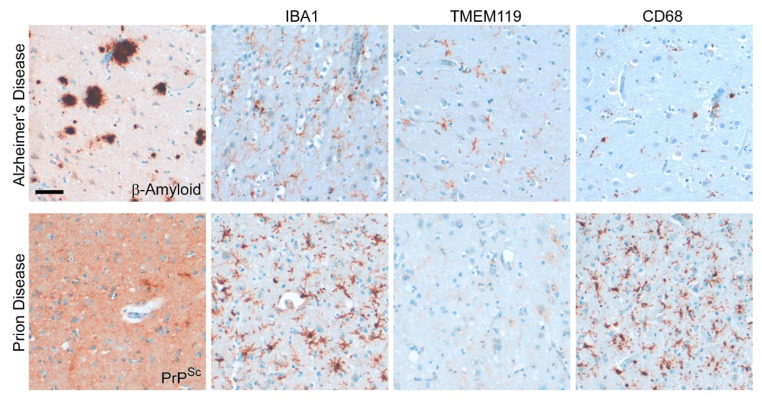
**Disease-associated microglia profile differs between human sCJD and AD.** Frontal cortex tissue of a sCJD patient and an age-matched AD patient show reactivity for the pan-microglia/macrophage marker IBA1, the microglial marker TMEM119, and the activation marker CD68 [97]. Deposits of misfolded proteins are shown for the respective antibodies against amyloid-β (AD) or PrP^Sc^ (sCJD). Note that the deposition pattern of misfolded PrP^Sc^ may vary considerably between CJD patients (see also Figure 6). In contrast to AD, microglia activation is more prominent and the microglial homeostasis marker is almost completely lost in sCJD [97]. Scale bar: 50 µm.

**Table 1 cells-11-02948-t001:** Summary of clinical parameters of Creutzfeldt–Jakob disease patients, an Alzheimer’s disease (AD) patient, and age- and gender-matched human post-mortem control brain samples displayed in this review. All cases underwent standardized neuropathological assessment, including macroscopic and microscopic examination. sCJD and AD diagnoses were neuropathologically confirmed according to current criteria. Controls did not show any sign of neurologic or neurodegenerative diseases [73,97].

Patient	Disease	Age	Gender	ABC Score	CJD Subtype	Cause of Death
1	sporadic CJD	78	M	—	MV2K+C	CJD
2	sporadic CJD	69	F	—	VV1	CJD
3	sporadic CJD	77	F	—	MM/MV1	CJD
4	Control	69	F	—	—	Bronchogenic adenocarcinoma
5	Control	79	M	—	—	Ventricular fibrillation
6	AD	80	M	A3B2C3	—	Dementia

## Data Availability

All data are described in this study.

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
