# Peer review of "Loss of Homeostatic Microglia Signature in Prion Diseases"

_cells, 2022, doi:10.3390/cells11192948_

Round 1
Reviewer 1 Report (Previous Reviewer 4)
The authors have addressed almost all of my concerns and I have no further comments.
Author Response
We thank the reviewer for this positive reply.
Reviewer 2 Report (Previous Reviewer 1)
The authors responded to my comments satisfactorily. Indeed, turning the manuscript into a review article was a good choice. I have only one final comment that, in my opinion, can be resolved during proof (if the article is accepted) or through minor revision: the names of human genes must be written in capital letters and in italics. Protein names must be capitalized only, without italics. This should be reviewed throughout the text.
Author Response
We thank the reviewer for this positive reply. We apologize for the mistake regarding the protein and gene names and corrected them accordingly.
Reviewer 3 Report (Previous Reviewer 3)
The authors have considered the major issues I raised concerning the original submission. However, despite the changes made this manuscript remains a hybrid between a review article and a research study. The authors have attempted to align their manuscript towards a review, but much apparently novel data remain within it. While the authors stress this is a review, there are important concerns with the inclusion of these data within the manuscript. The authors themselves highlight limitations with these data and suggest additional validations etc. However, my major concern is that when such preliminary data or unsupported statements are included in reviews they have the tendency to become dogma or accepted in the field, or used to refute other peer-reviewed studies, irrespective of any concerns or limitations with those data. This can cause important issues for others when submitting research manuscripts, proposals etc. that may/may not agree with the statements made. I therefore cannot support the inclusion of significant portions of previously unpublished data in this review manuscript. Reviews should be up-to-date at the time of submission/acceptance. A quick Pubmed search suggests there some important recent advances on the contributions of microglia in prion disease that are not discussed.
Author Response
We are sorry that the revised version of the manuscript was containing too much primary data. One of the reviewers requested novel data on P2RY12 staining in mouse and human brain tissue. We agree that this does not fit into a review and, accordingly, deleted it from the manuscript. However, when providing a revised manuscript, one has to address all (here 4 different) reviewers. Regarding the other figures, we rephrased the wording and made it more clear, where we reproduce figures from (our) published manuscripts. This manuscript does not contain preliminary data anymore. All the data shown in this manuscript have been published in peer-reviewed journals in slightly other versions. This is common practice for a review and we adapted our wording according to the review published in Cells (Cells 2021, 10(9), 2236; https://doi.org/10.3390/cells10092236 Beyond Activation: Characterizing Microglial Functional Phenotypes). All our data have been shown before by us and others in this or a slightly modified form. Moreover, M. Glatzel, H. Altmeppen, and I (S.Krasemann) have longstanding expertise in studying murine and human prion diseases or microglia biology (S.Krasemann). At the request of the above-mentioned reviewer, we established alternative antibodies for P2RY12 staining in brain tissues that showed similar results as the one displayed in this review but were now deleted from the revised version. Thus, we feel that this review is not proposing something unrealistic, but rather highlights valid findings of microglia dysregulation in prion pathophysiology. Moreover, we made it more clear, where we cite our own data so that the readers can make up their own opinion. We hope that we can convince this reviewer with the revised version of our manuscript.
We are very sorry that we did not meet the expectations of the reviewer regarding the selection of the literature. We included nine additional novel manuscripts now, some of which had not been released at the time of initial submission. With this, we hope to meet the expectations of the reviewer.
Reviewer 4 Report (Previous Reviewer 2)
The reformatted manuscript addressed most of my points of concern.
Minor point
Line 1: “Article” should be changed to “Review”.
Author Response
We thank the reviewer for this positive reply. We apologize for the mistake and corrected it accordingly.
Round 2
Reviewer 3 Report (Previous Reviewer 3)
I appreciate the consideration and attention that the authors gave to my concerns. The manuscript is much improved and now adequately conforms to a review style. Given the changes made I am happy to recommend acceptance.
This manuscript is a resubmission of an earlier submission. The following is a list of the peer review reports and author responses from that submission.
Round 1
Reviewer 1 Report
Wang et al. evidenced the loss of homeostatic microglia signature in prion diseases, based on data obtained with murine and human brain tissues. The study is well written, with an informative introduction (especially Figure 1) and discussion, showing publication merit especially because the study is rich in histopathological and cellular findings of prior diseases, something that is not simple to obtain. However, I have some minor points that could be improved by the authors:
- Lines 35-36: Add some information concerning the genetic aspects of sporadic CJD.
- Lines 69-74: (I) It is important to rewrite this section to make it clear to the reader that this is an original article, additionally containing a review on the subject. As it is currently written, it appears that this is a review article only, which is not the case. (II) I also recommend describing clearly the study objective in this paragraph.
- Lines 94-100: The number of animals used in the procedures is not clear. Please, correct it.
Reviewer 2 Report
In this work, Wang and colleagues overview the involvement of microglia in prion disease pathophysiology by focusing on their function. The authors summarize the current knowledge on the dysregulation of glial cells and the dual role of microglia in prion diseases. Through analyzing immunohistochemical staining of brain sections from prion-infected mice and sCJD patients, the authors demonstrated dysregulation of microglia and a loss of homeostatic microglia proteins, which are apparent as contrasted with those from AD patients. The manuscript is well-structured, written carefully, and easy to follow. I think this manuscript provides important insights into prion fields.
However, according to the instructions for authors of the journal, “Articles” are defined as “all original research manuscripts, provided that the work reports scientifically sound experiments and presents a substantial amount of new information.”. I am not convinced whether this manuscript fulfills the latter criteria since the loss of microglia homeostatic phenotype in prion disease was demonstrated solely by immunohistochemical analysis of marker proteins. In addition, some co-authors have already reported low expression of one of the marker proteins in brain tissues from sCJD patients (Krbot et al., 2018). It might be a good idea to switch article types from “Articles” to “Reviews”, if possible.
Minor points
Line 96: What is the difference between RML and RML 5.0?
Line 227: IL 34 is not a microglia receptor and IL34-deficient mice contain fewer microglia than wild-type mice.
Line 329: Upregulation of P2ry12 in primary microglia isolated from 22L-infected mice was reported (Sinha A, et al., 2021). Please discuss the discrepancy between results from the current and the previous study.
Line 445: For the uninformed reader, please define “YKL-40”.
Figure 2: For clarity, pink arrowheads should be replaced with brightly colored arrowheads such as white color.
Reviewer 3 Report
Here the authors describe/discuss the concept that prion disease in the CNS is accompanied by the loss of a homeostatic signature. The manuscript is well written but unfortunately to this reviewer it was uncertain whether I was reading a review or a research paper. If this is a review then it should not contain novel research data. If this is a research paper it should describe a novel set stand alone set of experiments. If I understand correctly, this is a research paper. Then it is my opinion that there is only limited novel data presented which do not provide the significant novelty or importance for publication in the Journal. These data are restricted to just the last few figures only. The majority of the figures present data that has been presented in many similar studies.
The authors report that dysregulation of the microglia is evident by d 70-80 (Figure 2). However the supporting data simply show increased IBA1 staining. This does not on its own confirm dysregulation and could equally be part of a tightly regulated response of the microglia to prion infection. The authors data supporting their conclusion that the microglia signature is lost as a consequence of prion infection is not a novel concept and has been reported in a variety of independent studies. Importantly, in the current manuscript this conclusion is based on the limited analysis of a very small number of microglia markers by immunostaining. The reduction in the expression of these markers reported in the current study appears to be in contrast to data from a range of transcriptional studies showing increased expression of the same markers as disease progresses.
Reviewer 4 Report
In the present study, Wang et al. studied microglial activation and its phenotype in prion diseases. It is already well known that both neuronal death and associated glial activation occur in the late stages of prion diseases, so some of the results in the first half of this paper are not necessarily novel. However, they claim that microglia are highly dysregulated in prion diseases and that the loss of microglial-specific homeostatic markers including P2RY12 and TMEM119 stands out. The authors should explore this interesting point a bit more, otherwise this paper would be descriptive in its current state and lack a clear mechanistic insight. Although the results are potentially interesting, I have several points that should be addressed by the authors. Specific comments are as follows.
Major points.
1. Conclusions based on microscopy images for homeostatic markers should be backed up by quantitative analyses. Also, the loss of homeostatic markers should be confirmed by western blotting.
2. There is only one example of Alzheimer’s Disease (AD) and the authors should increase the number of samples.
3. The study lacks physiological significance of the loss of homeostatic markers. How do microglia that have lost homeostatic markers differ from the original cells? Are they M1-like or M2-like cells? What are the implications for non-cell autonomous microglia-mediated neuronal death? What are the consequences for prion diseases? Addressing these mechanistic questions would significantly strengthen the manuscript.
Minor points.
1. It would be better to detail which specific part of the brain region. e.g. What part of the hippocampus? What part of the thalamus?
2. Figure 5B: Why is there little expression of GFAP at thalamus in the healthy brain?
3. Figure 5, 6: The authors should also present the data for P2RY12.